# New Insights in *bla*_KPC_ Gene Mobilization in *Pseudomonas aeruginosa*: Acquisition of *bla*_KPC-3_ and Identification of a New Tn*2*-like NTE Mobilizing *bla*_KPC-2_

**DOI:** 10.3390/antibiotics14090947

**Published:** 2025-09-19

**Authors:** Deisy Abril, Juan Bravo-Ojeda, Julio-Cesar Garcia, Aura Lucia Leal-Castro, Carlos Humberto Saavedra-Trujillo, Johana Madroñero, Rosa-Helena Bustos, Ricaurte Alejandro Marquez-Ortiz, Zayda Lorena Corredor Rozo, Natasha Vanegas Gómez, Javier Escobar-Pérez

**Affiliations:** 1Bacterial Molecular Genetics Laboratory-LGMB, Universidad El Bosque, Ak. 9 #131a-02, Bogota 110121, Colombia; djabril@unbosque.edu.co (D.A.); rmarquez@unbosque.edu.co (R.A.M.-O.); zcorredor@unbosque.edu.co (Z.L.C.R.); natashavanegas@yahoo.es (N.V.G.); 2Facultad de Medicina y Grupo de Investigación en Enfermedades Infecciosas, Universidad Nacional de Colombia, Bogotá 110111, Colombia; jbravoo@unal.edu.co (J.B.-O.); allealc@unal.edu.co (A.L.L.-C.); lmadronero@unbosque.edu.co (J.M.); 3Evidence-Based Therapeutics Group, Department of Clinical Pharmacology, Faculty of Medicine, Universidad de La Sabana and Clínica Universidad de La Sabana, Campus del Puente del Común, Km. 7, Autopista Norte de Bogotá, Chía 140013, Colombia; rosa.bustos@unisabana.edu.co; 4Facultad de Medicina, Universidad Nacional de Colombia, Bogotá 110111, Colombia; chsaavedrat@unal.edu.co; 5The i3 Institute, Faculty of Science University of Technology, Sydney 1168, Australia

**Keywords:** *Pseudomonas aeruginosa*, *bla*
_KPC_, carbapenem resistance, Tn*4401b*, NTE_KPC_-IIg

## Abstract

Carbapenem-resistant *Pseudomonas aeruginosa* is a major cause of healthcare associated infections in hospitalized patients and what is more warring with reduced therapeutic options. The KPC is a powerful enzyme capable of hydrolyzing the carbapenems, described first in *Klebsiella pneumoniae* and it already has found in *P. aeruginosa.*
**Objective**: To perform a comparative genomic analysis of two new genetic platforms mobilizing the *bla*_KPC-2_ and *bla*_KPC-3_ in two ST111 and ST235 pandemic clones of *P. aeruginosa* in Colombia, South America. **Methods**: Sixty-six *bla*_KPC_-harboring *P. aeruginosa* isolates were identified and characterized during a prospective study conducted in six high complex hospitals in Colombia. Genetic platforms mobilizing the *bla*_KPC_ were analyzed. **Results**: The *bla*_KPC-2_ and *bla*_KPC-3_ were identified in 24 and 42 isolates, respectively. The *bla*_KPC-2_-harboring isolates belonged to ST235 and *bla*_KPC-3_ to ST111. The whole genome sequencing indicated that the *bla*_KPC-3_ gene was mobilized by the Tn*4401b* within a 55-kb-size environmental origin plasmid, which, in other isolates, was inserted into the chromosome through a transposition event of IS*Pa38*. Regarding the *bla*_KPC-2_ gene, this was mobilized by a new Non-Tn*4401* Element (NTE) derived from transposon Tn*2* (proposed as variant IIg), which has been transposed into a 43-Kb-size little-studied plasmid related to *Klebsiella* spp. **Conclusions**: Our results reveal a new acquisition event of *bla*_KPC_ in *P. aeruginosa,* in this case *bla*_KPC-3_. Likewise, the pandemic high-risk clones ST111 and ST235 of *P. aeruginosa* continues to spread *bla*_KPC_ gene through different mobile genetic elements, jumping of conventional Tn*4401b* and acquiring new Tn*2*-derived NTE, which were inserted in diverse plasmids.

## 1. Introduction

*Pseudomonas aeruginosa* is one of the most versatile bacteria, which lives mainly in the environment but evolutionarily has acquired and stabilized molecular mechanisms to survive into the human body yielding diverse and potentially mortal infections. According to the World Health Organization (WHO), *P. aeruginosa* is included in the “critical” category and is an important cause of related infections, mainly in intensive care units (ICU). Infections caused by this microorganism are associated with high morbidity and mortality. Initially, these infections were successfully controlled by the usage of antibiotics, however, *P. aeruginosa* has achieved to overcome it due to the emergence the different and potent resistance mechanisms such as deregulation of constitutive genes (i.e porins and efflux pump), acquisition of many resistance genes, modification of cell wall, alteration of drug target, degradation/alteration of drug, biofilm formation [1]. One of these powerful mechanisms is the carbapenemases, enzymes capable of degrading carbapenems (and almost all the β-lactam antibiotics), molecules that have been the most effective against *P. aeruginosa* infections. Unfortunately, the resistance rate to carbapenems has been increasing worldwide, reaching levels up to 93,5% (reported in Ukraine) [2]. Several carbapenemases have been identified in *P. aeruginosa* such as VIM, IMP, SMP, NDM, OXA-48, GIM, CAM, FIM, AIM, HMB, and KPC [1].

The KPC was first identified in a *K. pneumoniae* isolate from United States of America in 1996 and since then it has already been reported in countries of the five continents [3]. The KPC enzyme in *P. aeruginosa* was initially reported from Colombia in 2007 [4], and since then has increased its detection in other countries in America, Asia and Europe [5,6,7,8,9,10,11]. In Colombia, the carbapenem resistance in *P. aeruginosa* has been increasing during the last years up to 9.1% [2], whereby well-structured surveillance studies are essential to establish the behavior and impact of the arrival of these “new” genes on the infections caused by *P. aeruginosa*. Dissemination of *bla*_KPC_ in this species has been associated initially by Tn*4401b* (mobile genetic element (MGE) most worldwide disseminated in *Klebsiella pneumoniae*) and NTE_KPC_-IIc [12]. Over the course of time, the appearance of new genetic arrangements with more remnants of Tn*4401b* (NTE) have been circulating on different plasmids in both species, some of them in new transposon structures, making the study of mobilization of *bla*_KPC_ increasingly most important [13,14]. In this way, we conducted a genetic and genome surveillance prospective study of the carbapenem-resistant *P. aeruginosa* isolates recovered from hospital infections in patients, and we achieved to detect: first, a new plasmid mobilizing the *bla*_KPC-2_ into a genetic platform derived from Tn*2* transposon circulating within isolates belonging to the high-risk pandemic clone ST235, and second, the *bla*_KPC-3_ acquisition by the high-risk pandemic clone ST111 in both plasmid and chromosome (stabilization process) through a transposition event of the insertion sequence IS*Pa38* (or Tn*Pa38* as it is proposed by this study).

## 2. Results

### 2.1. The bla_KPC_-Harboring P. aeruginosa Isolates Increase Their Frequency in the Colombian Hospital Settings

During the period of study, 128 carbapenem-resistant *P. aeruginosa* isolates were identified from 121 patients who attended six health institutions in Colombia. The susceptibility profiles of the isolates are showed in Appendix A. To highlight, 100 (78.7%) isolates were resistant to the three or more families of antibiotics evaluated (β-lactams, quinolones, aminoglycoside and polymyxin), it significantly reduced the therapeutic options. Considering the importance of carbapenem resistance, 66 (51.6%) and 54 (42.2%) isolates harbored *bla*_KPC_ and *bla*_VIM_, respectively; however, the co-existence of *bla*_KPC_ and *bla*_VIM_ was observed in 35 (64.8%) of these, being the dual presence of these carbapenemases in Colombian *P. aeruginosa* isolates becoming increasingly frequent.

### 2.2. The Pandemic High-Risk Clones ST111 and ST235 Continue Being the Catcher of the bla_KPC_ Gene

The *bla*_KPC-2_ has been the principal variant reported in *P. aeruginosa* in Colombia and the more frequent in the world*,* we determined the *bla*_KPC_ variant of routine in the isolates, hoping that were variant 2; however, surprisingly we found that 42 (63.6%) isolates harbored the variant 3. Now, among the 11 different pulsotypes found by PFGE, the *bla*_KPC-3_ gene was associated with pulsotypes a, b, and c; and notably, these all belonged to the ST111, like others reports with this KPC variant in Colombia [15]. The remain eight pulsotypes had the *bla*_KPC-2_ gene and belonged to ST235, as has previously been reported for others *P. aeruginosa* isolates [16].

### 2.3. The bla_KPC_ Gene Is Being Mobilized for New and Different Genetic Platforms

To determine the genetic platforms mobilizing the *bla*_KPC_ gene, we used the WGS to establish the chromosome and plasmid sequences of four representative isolates selected on their frequency, *bla*_KPC_ variant, PFGE pulsotype and ST. The isolates sequenced were 30Pae2 (ST111), 34Pae8 (ST235), 34Pae23 (ST235) and 34Pae36 (ST111). 30Pae2 had a circular chromosome of 7.2 Mbp and harbored the plasmid p30Pae2 of 55.6 Kbp, this isolate had a resistome of 14 genes, 13 of them (*bla*_PDC_, *bla*_TEM_, *bla*_OXA-395_, *bla*_VIM-2_, *sul1*, *aac(3)-IIa*, *aac(6′)-29*, *aac(6′)Ian*, *aadA13*, *aadA6, aph(3″)-lb*, *aph(3′)-IIb* and *aph-(6)-Id*) were in the chromosome and *bla*_KPC-3_ in p30Pae2-KPC. 34Pae8 had a chromosome of 6.8 Kbp and the plasmid p34Pae8-KPC of 42.7 Kbp, from this isolate 10 genes were identified, *bla*_PDC_, *bla*_OXA-488_, *catB7*, *sul1*, *aph(3″)-IIb*, *aadA6*, *qacEΔ1*, *fosA* were in chromosome, and *aac(6′)-29* and *bla*_KPC-2_ in p34Pae8-KPC. 34Pae23 had a partial chromosome of 6.8 Kbp and a plasmid p34Pae23 of 42.9 Kbp, this isolate had the same resistance genes as 34Pae8. 34Pae36 had a complete chromosome of 7.4 Kbp and a plasmid p34Pae36, for this isolate 10 genes were identified only in the chromosome: *bla*_PDC_, *bla*_OXA-395_, *bla*_KPC-3_, *bla*_VIM-2_, *catB7*, *sul1*, *aac(6′)-29*, *aph(3″)-IIb*, *qacEΔ1* and *fosA* (Appendix A).

In the two isolates belonging to ST235, the *bla*_KPC-2_ gene was transported in an almost identical plasmid (99.76% identity), p34Pae8-KPC had a 248 bp deletion, compared to p34Pae23-KPC which had a size of 42,977 bp, corresponding to a genes region of a hypothetical protein and a recombinase (Figure 1), does not belong to any incompatibility group previously reported, does not possess additional resistance genes, and the characteristic Tn*4401* was not found.

An analysis of the *bla*_KPC-2_ gene surroundings showed the presence of a variant of the transposon Tn*2* (upstream) with Δ*bla*_TEM_ gene, which has embedded one copy of an insertion sequence with remarkable similarity (99% of identity) with the recently reported IS*Kmi1* (Figure 2A). This IS belongs to the IS*30* family and was first detected in *K. michiganensis*, an emerging pathogen first reported in 2013 [17] that posteriorly become carbapenem-resistant due to the acquisition of the *bla*_KPC-3_; and also in *Enterobacter hormaechei* (GenBank accession number NZ_CP049047) [18]. In addition, the IS*Kmi1* insertion occurred in an independent transposition event (respect with Tn*2* transposition) that yielded a 2-bp size target site duplication (TSD), characteristic size duplication of the IS*30*-related transposases. This insertion also altered the promoter region of the *bla*_KPC_ gene by introducing a new “-35” promoter element within the right-hand terminal inverted repeat (IRR), as has previously been demonstrated for IS*30* elements [19]. This modification could potentially affect transcription of this gene, although further experimental validation is required (Figure 1 and Figure 2B,C). 

No additional mobile genetic platforms were identified within the plasmid. Although the Tn*2* variant was initially suspected to mediate *bla*_KPC_ gene mobilization, its left IR was found precisely 74 bp upstream of the gene, suggesting that it may not play a direct role in the transposition of *bla*_KPC_. 

A deeper analysis of the Tn*2* surroundings allowed found traces related with the transposon Tn*As1* (11% of coverage and 85.6% of identity), including their IRs, which are located exactly to 893 bp downstream of the Tn*2* and 886 bp upstream of *bla*_KPC_ (Figure 2A). Notably, these IRs have an 86% of identity respect with the IRs of the transposon Tn*2* and have identical region at the 3′ site (GTTTTC) and mainly at the 5′ cut recognition site for transposase (GGGG) (Figure 2B).

This plasmid sequence has 99% of identity and 99,8% of query cover with a plasmid unnamed of *P. aeruginosa* strain HdC (42,750 bp); however, although this plasmid was recently reported, a deep analysis of the mobilization structures was not performed and therefore Tn*2*-NTE_KPC_-IIg was not identified [20]. Another interesting feature was found when the plasmid p34Pae23-KPC was compared with a *bla*_KPC_-negative plasmid (unnamed) identified within the *P. aeruginosa* strain AR441 (GenBank accession number CP029092), it seems to show a probable insertion of this Tn*As1* IRs-delimited DNA fragment just within the *stbC* gene (without to affect its open reading frame), and whose transposition would have yielded a 5-bp size TSD (TAGAA) (Figure 3). Taking together these results, we hypothesize that the transposase of the Tn*2* could have used the *TnAs*1 IRs as an alternative route in the recognizing and transposition process of DNA and thus participate in the *bla*_KPC_ mobilization. Additionally, this Tn*2*-based *TnAs*1 IRs-delimited structure corresponds to NTE_KPC_ elements, that, for the Δ*bla*_TEM_ presence upstream of *bla*_KPC_ and ΔIS*Kpn6* downstream (distinctive fact for the members of this group) would belong to the group II but being a new isoform. The genetic structure of this new NTE_KPC_-II is not related to any of the previously reported subgroups (a–f) and we propose that this structure can be described as a new subgroup NTE_KPC_-IIg (Figure 2D). When looking for NTE_KPC_-IIg of p34Pae8-KPC (hospital A) and p34Pae23-KPC (hospital D) in all positive *bla*_KPC_ isolates, it was found in 18 (27.3%) isolates from the five hospitals, suggesting that there was a wide dissemination of this plasmid in the city. Now, a genetic structure related to this NTE_KPC-_IIg lies within the plasmid pKP18-50-tet(A) (GenBank accession number: MN268581), which was recovered from *Klebsiella pneumoniae* in 2018 and it seems to be associated to the dissemination of the *tet* genes in China.

Regarding the ST111 isolates, the sequencing revealed that the *bla*_KPC-3_ gene is being mobilized for other platforms. In the case of the isolate 34Pae36, one unique contig, with a size of 7,418,292 bp, was found indicating que the *bla*_KPC-3_ was chromosomally located; by contrast, the genome of the isolate 30Pae2 was formed for the chromosome (7213.418 bp) and one plasmid (named as p30Pae2-KPC) with a size of 55,696 bp and where the *bla*_KPC-3_ gene was located. This plasmid does not belong to any incompatibility group previously reported. Unlike the *bla*_KPC-2_ harboring isolates, the *bla*_KPC-3_ was mobilized for the transposon Tn*4401* isoform b in both isolates (30Pae2 and 34Pae36). The search for p30Pae2-KPC with Tn*4401b* in *P. aeruginosa* isolates harboring *bla*_KPC_ revealed that this plasmid was found in 33 (50.0%) isolates, of which 32 came from the institution F (where isolate 30Pae2 was recovered) and the remaining isolate from the institution C (where 34Pae36 was also found), so the dissemination of this plasmid seems to be associated to an institutional level than a regional. Interestingly, the insertion of this plasmid in the chromosome region of 34Pae36 was not observed in any other isolate.

A standout fact was that the Tn*4401b* flanking sequences were not identical, that is, a TSD was not yielded; but these flanking sequences were the same in both structures (plasmid and chromosome), suggesting a possible shared mobile platform. The comparative genomic analysis displayed the insertion of an almost identical plasmid sequence (100% of identity and 96% coverage) into the chromosome (Figure 1). These results show that the chromosomal *bla*_KPC-3_ insertion in the isolate 34Pae36 was not produced for the Tn*4401b* transposition, as it was previously reported by us in other Colombian isolates [16], but for plasmid insertion. Now, what possible genetic mechanism could be involved?

### 2.4. ISPa38/TnPa38, a New Player Moving Resistance Genes in P. aeruginosa

A detailed analysis of the chromosomal site where one copy of the plasmid p30Pae2-KPC was inserted within the isolate 34Pae36 exhibited the presence of the two directly oriented copies of the IS*Pa38,* abutting the plasmid sequence and forming a pseudo-composite transposon structure (Figure 4B). The IS*Pa38* has a size of 6455 bp, is bounded by the terminal inverted repeats of 39 bp, and is constituted of eight genes (transposase, resolvase, and six-passenger genes of unknown function) and we achieved to identify a putative *res* site. It is more appropriate to refer to this structure as a transposon (Tn*Pa38*) rather than an IS. The TnpA protein of IS*Pa38*/Tn*Pa38* contains the DDE triad within its catalytic domain and belongs to the Tn*3* family or pMULT-4 family (prokaryotic Mutator-Like Transposase) according to the classification proposed by Guerillot et al. [21]. Notably, in the isolate 30Pae2 (that harbored the plasmid p30Pae2-KPC) only one copy of the IS*Pa38*/Tn*Pa38* was found within its chromosome, it was justly inserted in the intergenic region of two genes encoding a hypothetical protein, and a truncated transposase, respectively. This suggests that the chromosomal insertion of the plasmid was accompanied by a duplication of this IS.

To try to decipher the possible transposition events involved in the formation of this structure, we performed an analysis of the IS*Pa38*/Tn*Pa38* flanking sequences (FS). In the 30Pae2, the IS*Pa38*/Tn*Pa38* was flanked by sequences ATGAA and TTCCA (no TSD produced), that is, probably its mobilization was not yielded by a “copy and paste” transposition (Figure 1 and Figure 4B), as it has been described in Tn*3* [22]. In this same way, in the 34Pae36, the pseudo-composite transposon structure was also flanked by these two sequences, but interestingly, a TSD was generated in the internal sequences (AGTTC) (Figure 1 and Figure 4B). These genetic transposition prints suggest that the structure within the 34Pae36 chromosome corresponds to a cointegrate (chromosome-plasmid), which was probably yielded by a replicative transposition event of the IS*Pa38*/Tn*Pa38* from chromosome toward plasmid [22].

These results indicate that IS*Pa38*/Tn*Pa38* is involved in the mobilization (and possible stabilization) of the *bla*_KPC_ gene toward the chromosome of *P. aeruginosa*. In addition, the formation of this IS*Pa38*/Tn*Pa38*-pseudo-composite transposon could have implications in the *bla*_KPC_ dispersion because it could facilitate the formation of new TU structures, circular structures generated by the resolution of the cointegrate for either mechanism IS*Pa38*/Tn*Pa38*-TnpA depend or for homologous recombination (RecA), and the possible mobilization as a composite transposon, as it has been described in Tn*3* [22]. However, more studies oriented to establish the functionality and transposition routes used for IS*Pa38*/Tn*Pa38* are necessary.

Then, how could *P. aeruginosa* have acquired the IS*Pa38*/Tn*Pa38*? From our analysis there are several points to highlight, first, the IS*Pa38*/Tn*Pa38* is almost exclusive to *P. aeruginosa*; when we examined the frequency of this IS within *Pseudomonas* spp., we found that it has only been described in *P. aeruginosa* strains (except for the clinical *P. putida* strain H8234) [23]. Second, the IS*Pa38*/Tn*Pa38* is scarce in *P. aeruginosa,* from 5678 genomes publicly available from NCBI (last accession November 2021), this IS was only found in 13 genomes. An analysis of the IS*Pa38*/Tn*Pa38* surroundings and FS revealed that this IS has been acquired by at least three independent genetic events, all directly toward the chromosome (Table 1). An acquisition event was produced at the position ~ 6,200,000 (just at the intergenic region of IPC90_06220 and truncated *tnpA* genes), where the IS is immersed in a genomic island region of 106,730 bp within a probable prophage with some regions genetically related with the filamentous bacteriophage Pf1 (AY324828) with 92.6% of identity and 6% of coverage, and was inserted within a *tRNA^Met^* gene copy (Figure 5). Interestingly, this insertion was found exclusively in strains previously reported belonged to ST111 (included the 34Pae36 isolate). Another IS*Pa38*/Tn*Pa38* insertion was located at site 4,250,295 exclusively within a genomic island in the strain AR_0440 (ST357). Finally, a third insertion was produced at the site ~3,000,000 and perhaps due to the acquisition of a pathogenicity island for diverse genetic background (ST175 and ST155), included the well-studied strain DK2 (ST386). No TSD were found in all IS*Pae38*/Tn*Pa38* copies (Table 1).

Due to little information about the IS*Pa38*/Tn*Pa38* in *P. aeruginosa*, we searched IS-related within the *Enterobacterales* genomes published in NCBI Nucleotide database, and we accomplished to identify a high similar IS (96% of coverage and 99% of identity) within the *Klebsiella pneumoniae* strains (Figure 6). Specifically, one IS*Pa38*/Tn*Pa38*-like copy was located within the plasmids pKPN_CZ, INF341-sc-2280155-plasmid 2, FDAARGOS1306, which have been associated with *qnrB* genes dissemination in Czech Republic and Australia [16]. By contrast to that found in *P. aeruginosa*, these IS*Pa38*/Tn*Pa38*-like copies are directly bounded by TSDs, all with a size of 5pb (Figure 6).

## 3. Discussion

The recommendations of the WHO over the last two decades to produce effective antibiotics with fewer side effects have borne fruit. Current guidelines from different scientific societies recommend antibiotics recently introduced to the market in the United States, such as cefiderocol, eravacycline, imipenem-cilastatin relebactam, meropenem vaborbactam, temamycin, and ceftazidime avibactam (CZA) [24,25]. Specifically, in difficult-to-treat resistant-*Pseudomonas aeruginosa* (DTR-PA), the guidelines include imipenem-cilastatin, relebactam, termomycin, ceftolozane/tazobactam, cefiderocol, and ceftazidime/avibactam. Unfortunately, in Colombia, as in other countries in the region, some of these therapeutic options are only found in major cities, which implies the use of non-recommended antibiotics such as polymyxins, fosfomycin, amikacin, gentamicin, and combination with high dose meropenem or doripenem. Given the limitation of therapeutic schemes for the treatment of infections caused by (DTR PA), it is important to highlight the urgent need to reduce antibiotic pressure in different hospital environments that, through the indiscriminate use of antimicrobials, promote the expression of resistance mechanisms such as those described in this article, added to others such as outer membrane porins (OprD), hyperproduction of AmpC enzymes, upregulation of efflux pumps, and mutations in penicillin-binding protein targets. Additionally, although CZA has shown adequate effectiveness against KPC-2-producing strains, reports of CZA resistance are beginning to appear, such as those that express *bla*_KPC-90_, whose genes have mutated from *bla*_KPC_ [26]. Studies of different mechanisms of resistance, availability, and access to the different available antimicrobials allow the generation of treatment guidelines for Latin American countries in accordance with our reality and force us to make a priority call for the introduction of new therapeutic alternatives in our pharmaceutical market to reduce morbidity and mortality by DTR PA [27].

Since 2008, we have carried out surveillance studies on resistance in *P. aeruginosa* in Colombia, what have allowed us to decipher the mechanisms adopted by this bacterium to overcome the antibiotic mortal pressure (especially to carbapenem) during these years. First, altering the expression of the constitutive genes such as deregulation of the porins, overexpression of some efflux pump, second, the acquisition of atypical class I integrons, inactivation of the genes encoding porins by IS*26* transposition, and third, mobile genetic elements like transposons, and plasmids harboring multiple resistance genes that have allowed the increase in the appearance of carbapenem-resistant and MDR *P. aeruginosa* isolates, being this last elements in our study, the most frequent (86%) [28].

The population of *P. aeruginosa* is composed of a small number of widespread clones, being the high-risk MDR clones ST235 and ST111 are the most worldwide observed harboring an arsenal or resistant genes [29,30]. These STs have already been previously reported with *bla*_KPC_ gene in Colombia [4,16,31,32,33] and other countries such as Brazil and Germany [10,11]. For the first time in the world, the *bla*_KPC_ gene was acquired by *P. aeruginosa* in Colombian isolates(4) through the two new and no related plasmids, the pPA2 (ST1006) first plasmid with an NTE_KPC_ in *P. aeruginosa*) and pCOL-1 (ST308) (GenBank accession numbers: KC609322 and KC609323.). The last one, has a size of the 31 kb and belong to the incompatibility group IncP-6 harboring *bla*_KPC_ in a Tn*4401b*. Interestingly, in some surveillance studies we have detected the plasmid pCOL-1 in *Klebsiella pneumoniae* isolates (where KPC is most frequent) suggesting a possible interspecies transference of this important resistance mechanism [34]. Since then, Tn*4401* was the most frequent transposon that have been mobilizing *bla*_KPC_ (almost all the variant 2) in ST235 and other STs and it has been mobilized in *P. aeruginosa* isolated from Colombia —even the double transposition of Tn*4401* in the ST235 chromosome of this species has been reported— IncHI1 plasmid in Germany and other plasmids not typeable by any known incompatibility group [16,35]. Being these last plasmids found recently harboring NTE_KPC_ in Brazil, Argentina and in this study with the new report of the structure Tn*2*-NTE_KPC_-IIg in p34Pae8-KPC and p34Pae23-KPC [10,20].

Although ST235 has been the most widely distributed clone worldwide, in Colombia ST111 harboring the *bla*_VIM_ gene was first reported —there are reports of the circulation of this carbapenemase gene in *P. aeruginosa* since 1999 without MLST results— arrived and became the most prevalent in infections produced by carbapenem-resistant *P. aeruginosa* isolates, as it happened in other countries with this ST and other pandemic clones such as ST175, 233, 277, 357, 654, and 773 [32,36,37,38,39]. ST111 co-harboring *bla*_VIM_ and *bla*_KPC_ genes have been reported since 2012, and that continues to be a problem of interest in our country, where overtime reports of this cocirculation have increased due to the increased clinical usage of carbapenems, as evidenced in this study, where 64.8% of the isolates presented this phenomenon for carbapenem-resistance [9,32,33,40,41]. While the dissemination of *bla*_VIM_ has been attributed in ST111 to its integration as a cassette gene in integrons, *bla*_KPC-2_ has been acquired by Tn*4401* and now, according to our results, is the MGE associated with *P. aeruginosa bla*_KPC-3_ acquisition [9,31,35].

In the present study, our results show that, in the six health institutions, the *bla*_KPC_ isolates have displaced the *bla*_VIM_ isolates and are already the most prevalent, suggesting a successful adaption of this resistance mechanism to this non-fermenting bacterium in our country. The *bla*_KPC_ gene has become the predominant carbapenemase-encoding gene in Colombia, largely due to its efficient dissemination through mobile genetic elements. This increasing frequency of KPC-producing *P. aeruginosa* isolates can be related with the mobilization of the *bla*_KPC_ gene toward new plasmid backbones, some of them, with an environmental origin; coming from *bla*_KPC_-positive plasmids circulating in *Klebsiella pneumoniae*, where this gene circulates with very high frequency and within diverse and very variables genetic platforms (including MGE and plasmids), these MGEs could be consulted at https://maphub.net/LGMB/KPC-Pseudomonas-aeruginosa-LGMB (accessed on 20 May 2025) [14,42].

Studies have shown that *bla*_KPC-2_ and *bla*_KPC-3_ are the most common variants, frequently detected in healthcare-associated infections (HAIs) and occasionally in community-acquired infections [15,43]. Regarding *bla*_KPC-3_, our findings were even more interesting, showing that this gene is been mobilized within a little-known plasmid structure, previously detected in isolates of environmental origin in Asia, some of these with *bla*_KPC_ incorporated, but variant 2, and mobilized for no classical Tn*4401* (Figure 2). In contrast, in the case of the plasmid p30Pae2-KPC, the Tn*4401b* was probably the first element responsible to mobilize the gene, but the insertion into the plasmid was helped maybe for another mobile structure (no TSD was found in Tn*4401b*). In this case, we also suppose the *bla*_KPC-3_ gene has arrived at *P. aeruginosa* from *K. pneumoniae* through the shared plasmids, due to that, in Colombia, the *bla*_KPC-3_ is the more frequently variant detected in *Klebsiella pneumoniae* isolates [44,45].

The overuse and misuse of carbapenems in Colombian healthcare settings have exerted strong selective pressure, favoring the emergence and dissemination of *bla*_KPC_-producing bacteria and in this case the establishment of *bla*_KPC_ in *P. aeruginosa.* Studies have shown that prior exposure to carbapenems is a significant risk factor for infections with carbapenem-resistant *Enterobacterales* (CRE) [33,46]. Deficiencies in infection control practices have played a pivotal role in the spread of *bla*_KPC_-producing bacteria in Colombian healthcare facilities. Horizontal gene transfer and clonal dissemination have been facilitated by inadequate hand hygiene, improper sterilization of medical devices, and insufficient isolation of infected patients [47]. These lapses have allowed *bla*_KPC_-producing isolates to thrive in hospital environments, contributing to their dominance over *bla*_VIM_-producing isolates. To reduce antibiotic pressure and thus slow the selection for resistance mechanisms in *P. aeruginosa* clinical isolates and other *Enterobacterales* species, it is essential that primary strategies can be implemented in hospitals (Appendix A) [48,49,50,51,52,53,54,55,56,57,58,59,60,61,62,63,64] and more genomic studies that allowing to know the mobile genetic elements with resistant determinants that are spreading in the institution, that allow a better dissemination control. The implementation of regional genomic surveillance programs has been shown to be effective in tracking the spread of *bla*_KPC_-harboring pathogens. For example, a study in three U.S. states demonstrated that genomic surveillance could identify distinct transmission pathways, such as importation versus clonal expansion, and inform targeted interventions [65] The clinical implementation of WGS has been shown to be feasible and effective for identifying transmission events of multidrug-resistant pathogens, including those harboring *bla*_KPC_. For example, a study in three Australian hospitals demonstrated that WGS-based surveillance can identify clusters of multidrug-resistant organisms and inform infection control interventions [66].

Standard and contact infection prevention and control practices, including hand hygiene, patient isolation, and environmental disinfection, remain critical for preventing the spread of *bla*_KPC_-harboring pathogens. Studies have shown that these practices, when implemented consistently, can reduce the transmission of multidrug-resistant organisms in healthcare settings [67].

## 4. Materials and Methods

### 4.1. Source of the Bacterial Isolates, Susceptibility Profiles and Molecular Detection of Carbapenemase Genes

The isolates were collected from an observational, descriptive, prospective, and multicentre surveillance study performed between 2017 and 2020 in six high complexity hospitals (designed as A–F) located in Bogota, the main city of Colombia. During this time, 128 carbapenem-resistant *P. aeruginosa* isolates were recovered from 121 patients, among those, 66 harbored the *bla*_KPC_ gene.

The susceptibility profile was determined using the automated systems VITEK^®^ 2 and the breakpoints defined by the Clinical and Laboratory Standards Institute according to the 2020 update. The MIC for meropenem was confirmed by microdilution test in these 30 carbapenem-resistant *bla*_KPC_-positive isolates.

The genomic DNA of isolates was extracted with DNeasy UltraClean Microbial Kit (QIAGEN N.V.) and presence of carbapenemases genes as *bla*_GES_, *bla*_IMP_, *bla*_VIM_, *bla*_OXA-48_ and *bla*_NDM_ was determined by PCR [68].

### 4.2. Genetic Relationship Establishment

Genetic relationship was determined by pulsed-field electrophoresis (PFGE) analysis, following the Centers for Disease Control and Prevention (CDC) protocol with minor modifications [69]*. P. aeruginosa* isolates were cultivated in BHI broth and incubated at 37 °C with shaking at 200 rpm for 12 h, to an OD625 of 1.5 in TE buffer (100 mM Tris + 100 mM EDTA, pH 8.0). 1% blocks were prepared with SeaKem^®^ Gold Agarose, with150 µL of bacterial suspension and 7.5 µL of proteinase K (20 mg/mL). For lysis, 1 mL of lysis buffer (50 mM Tris, 50 mM EDTA, pH 8.0 + 1% Sarcosyl) with 5 µL of proteinase K (20 mg/mL) was added to the blocks and incubated for 2 h at 55 °C at 175 rpm. Two washes were then performed with sterile deionized distilled water at 55 °C and two washes with TE buffer (10 mM Tris, 1 mM EDTA, pH 8.0). The blocks were stored at 4 °C until use. Restriction was performed with 12 units of the restriction enzyme SpeI (10 U/µL: Promega) in 1X Buffer B for 5 h at 37 °C. Restriction fragments were separated on a 1% SeaKem^®^ Gold Agarose gel in 0.5X TBE (Tris base; boric acid; 0.5M EDTA (pH 8.0); dH2O) buffer using the CHEFFII equipment (BioRad) with the following running conditions: initial time of 6.8 s and final time of 35.4 s for 23 h at an angle of 120° and a voltage of 6.0. The gels were stained with ethidium bromide (10 mg/mL). The image was captured with the ChemiDoc™ MP Imaging System (BioRad) and the dendrogram was obtained using the GelCompar II program (Applied Maths NV). For the analysis, *Salmonella* serotype Braenderup H9812 was used as a normalizing strain and the pulsotypes were defined according to the criteria of Tenover et al. with a cut-off point greater than 80% similarity [70].

For MLST analysis, fragments of the genes *acsA*, *aroE*, *guaA*, *mutL*, *nuoD*, *ppsA*, and *trpE* were amplified and sequenced according to the protocol previously reported [71]. The alleles and ST numbers were established from the MLST *P. aeruginosa* database (available from https://pubmlst.org/organisms/pseudomonas-aeruginosa, accessed on 20 May 2025).

### 4.3. Whole Genome Sequencing (WGS)

The genome sequencing was performed in four representative *P. aeruginosa* isolates, which were selected according to its PFGE pulsotype, ST, and the institution where they were recovered. Total DNA was extracted using the UltraClean^®^ Microbial DNA Isolation Kit (QIAGEN N.V). A 20 Kb-enriched SMRTbell™ library was prepared for each DNA sample. Libraries were sequenced using the PacBio RS II platform with the P6-C4 chemistry (one SMRT cell per sample). De novo assembly of subreads was performed with HGAP protocol in the SMRT Analysis v2.3. The contigs obtained were compared with each other for the presence of repeated ends, (suggestive of complete circular structures). Complete sequences were circularized as previously described [72]. Annotation of open reading frames was done in Prokka v1.14.5 [73] and the regions of interest were compared with Protein and Nucleotide NCBI databases using BLAST 2.16.0 and manually curated with ACT [74]. Antibiotic resistance genes were identified using ResFinder and CARD [75]. Genomic pairwise comparisons were plotted using Easyfig [76].

### 4.4. Comparative Genomics Analysis

A genomic comparative analysis was performed to determine, first, the similarities or differences in the platforms mobilizing the *bla*_KPC_ gene in the four Colombian *P. aeruginosa* isolates sequenced, second, comparison respect with the genomes published in the NCBI Nucleotide database, and third, to try to clarify the possible genetic mechanism involved in the *bla*_KPC_ mobilization.

Genomic and pathogenicity islands (GI and PI, respectively), prophages, insertion sequence (IS) elements and transposons were identified using IslandViewer [77], PHASTER [78], ISfinder (available at https://isfinder.biotoul.fr/, accessed on 20 May 2025) and the Tn Registry database (available at https://transposon.lstmed.ac.uk/tn-registry); respectively. The genetic platforms identified were manually curated throw ACT [21], and pairwise comparisons were visualized and illustrated and EasyFig [76].

### 4.5. Molecular Identification of bla_KPC_ Mobilization Platforms

Once the sequence of the mobile genetic elements associated with *bla*_KPC_ gene mobilization was determined, primers were designed to identify the insertion of Tn*4401* or NTE_KPC_ in the plasmid of interest, to know its frequency in the population of study. Multiplex PCRs was performed according to conditions described in Appendix A.

### 4.6. Accession Numbers of the Genome Sequences

The chromosome and plasmids sequenced established in this study have been deposited in the Nucleotide database (NCBI) with the accession numbers ON248939, CP095774, CP095775, CP095772, CP095773, and CP095770, which correspond to the sequences p30Pae2-KPC, 34Pae8, p34Pae8-KPC, 34Pae23, p34Pae23-KPC and 34Pae36; respectively.

### 4.7. Ethics Committee Approbation

This study was revised and approved by the Institutional Ethics Committee of the Universidad Nacional de Colombia (DIB-18-03). In addition, the study was approved by each participant institution. Experiments were performed in accordance with Declaration of Helsinki.

## 5. Conclusions

The appearance of *bla*_KPC_-transporting novelty plasmid backbones suggests that an active dispersion process of this resistance gene is taking place in *P. aeruginosa,* which is highly been facilitated for the conjunction of several genetic factors such as, first, the interspecies flux of plasmids from *Klebsiella pneumoniae* towards *P. aeruginosa*; second, the high transposition of the Tn*4401*; third, the emergence of new NTE_KPC_ structures, more compact and perhaps with a higher frequency of transposition; and forth, the participation of new transposons and ISs. For another hand, the pandemic high-risk clones ST111 and ST235 continue to demonstrate their genetic advantages with respect to other clones of *P. aeruginosa*, due to their very high ability to uptake and keep foreign DNA.

## Figures and Tables

**Figure 1 antibiotics-14-00947-f001:**
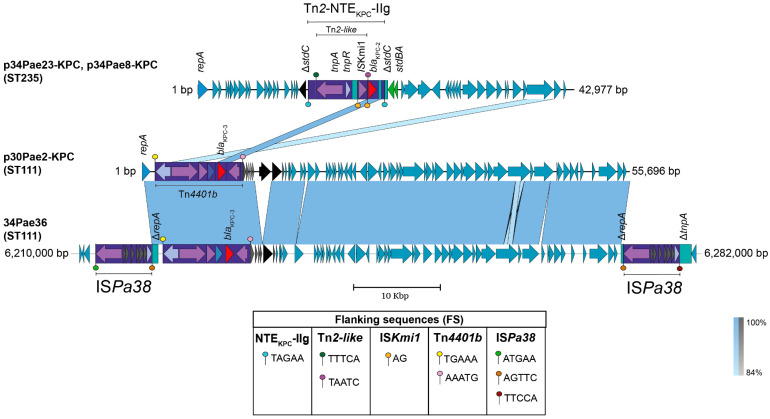
Comparison of the *bla*_KPC_ gene surroundings from four *Pseudomonas aeruginosa* strains sequenced p30Pae2-KPC (ON248939), 34Pae36 (CP095770), p34Pae23-KPC (CP095773) y p34Pae8-KPC (CP095775). The localization of the new Tn*2*-based NTE_KPC_–II element (isoform g) identified in the *P. aeruginosa* isolates p34Pae23-KPC and p34Pae8-KPC is showed. The shaded area between the sequences delimits the alignment regions with a percentage identity ≥ 84%. The arrows indicate the open reading frames (blue), transposases (purple), resolvases (lilac), and the *bla*_KPC_ gene (red). The ISs and transposons are highlighted and delimited by a purple rectangle. The flanking sequences (FS) for each mobile genetic element are displayed in the bottom box and are illustrated as color pallets. The graph has a scale line of 10,000 bp. ST: Sequence Typi.ng.

**Figure 2 antibiotics-14-00947-f002:**
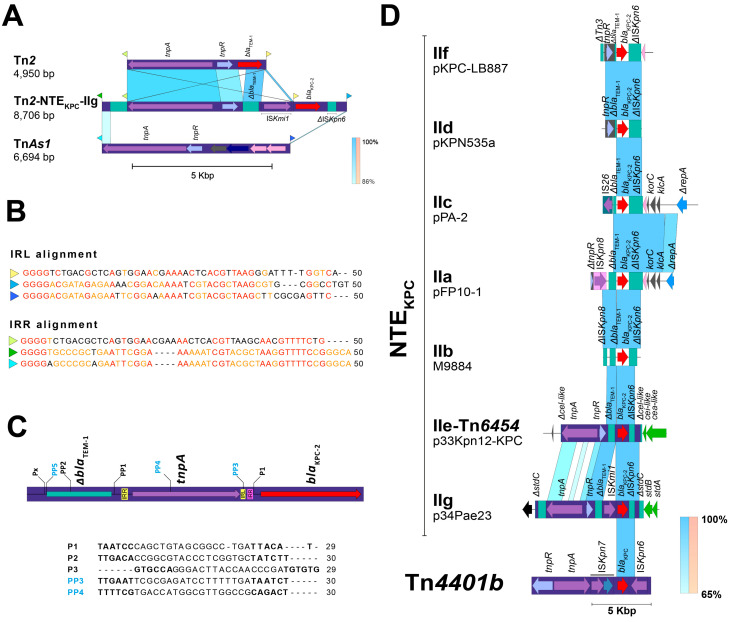
Analysis of the new Tn*2*-based NTE_KPC_–II element (isoform g) identified in the *Pseudomonas aeruginosa* isolate 34Pae8 and 34Pae23, (**A**) Genomic comparison of Tn*2* transposon and Tn*As1* with the new Tn*2*-NTE_KPC_-IIg. Structure of the new Tn*2*-based NTE_KPC_–IIg element and localization of the Tn*As1* traces. The flanking sequences (FS) for each mobile genetic element are illustrated as color flags. (**B**) Multiple alignment of the IRs belonging to Tn*2*, NTE_KPC_–IIg, and Tn*As1* showing the 5′ and 3′ conserved fragments (i.e., GGGG, TAAG and GTTTTC). (**C**) Identification of the two potential new promotors (PP3 and PP4) of the *bla*_KPC_ gene yielded for the upstream *IS*Kmi1 insertion and their comparison with promotor regions of Tn*4401:* P1 (Tn*4401a*, Tn*4401b*, Tn*4401f* and Tn*4401h*), P2 (Tn*4401a*, Tn*4401b*, Tn*4401c*, Tn*4401e*, Tn*4401f*, Tn*4401g*, Tn*4401h* and Tn*4401i*) y P3 (Tn*4401b* and Tn*4401f*) (**D**) Comparison with other NTE_KPC_-II elements (GenBank accession numbers: IIa, HQ651092; IIb, JN048641; IIc, KC609322; IId, MH595533; IIe, CP062794, IIf MT569433). The shaded area between the sequences delimits the alignment regions with a percentage identity ≥ 65. The arrows indicate the open reading frames (blue), transposases (purple), resolvases (lilac), and the *bla*_KPC_ gene (red). The ISs and transposons are highlighted and delimited by a purple rectangle. The graph has a scale line of 5000 bp.

**Figure 3 antibiotics-14-00947-f003:**
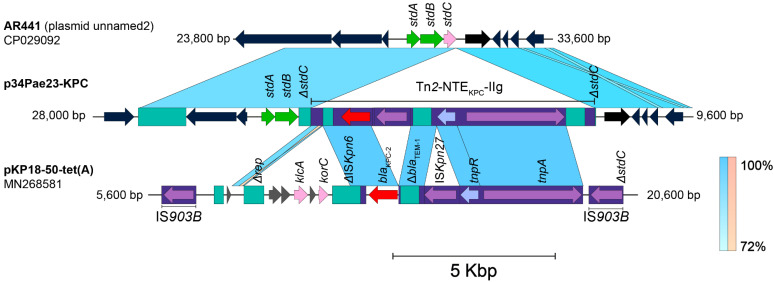
Identification of the putative site insertion of the Tn*2*-based NTE_KPC_–IIg element within the plasmid p34Pae23-KPC according to the comparison with the plasmid AR441 (Genbank accession number: CP029092). The shaded area between the sequences delimits the alignment regions with a percentage identity ≥ 72%. The arrows indicate the open reading frames (blue), transposases (purple), resolvases (lilac), and the *bla*_KPC_ gene (red). The ISs and transposons are highlighted and delimited by a purple rectangle. The graph has a scale line of 5000 bp. The comparative analysis of the plasmid p30Pae2-KPC revealed a genetic relation with seven poorly studied plasmids with and without the *bla*_KPC_ gene, which have been recovered from different species of *Pseudomonas* spp. (Figure 4A). To notice, the *bla*_KPC_-negative plasmids were isolated from *Pseudomonas* of environmental origin such as *P. putida.* It is plausible that these plasmids were circulating in the environmental setting and at some moment, were transferred to strains of *P. aeruginosa*, where, posteriorly, acquired the *bla*_KPC_ gene for different routes, first, through NTE structures (plasmids p14057-KPC, p1011-KPC2, and YLH6_p3) and then for conventional Tn*4401* transposon via, as happened with the plasmid p30Pae2-KPC.

**Figure 4 antibiotics-14-00947-f004:**
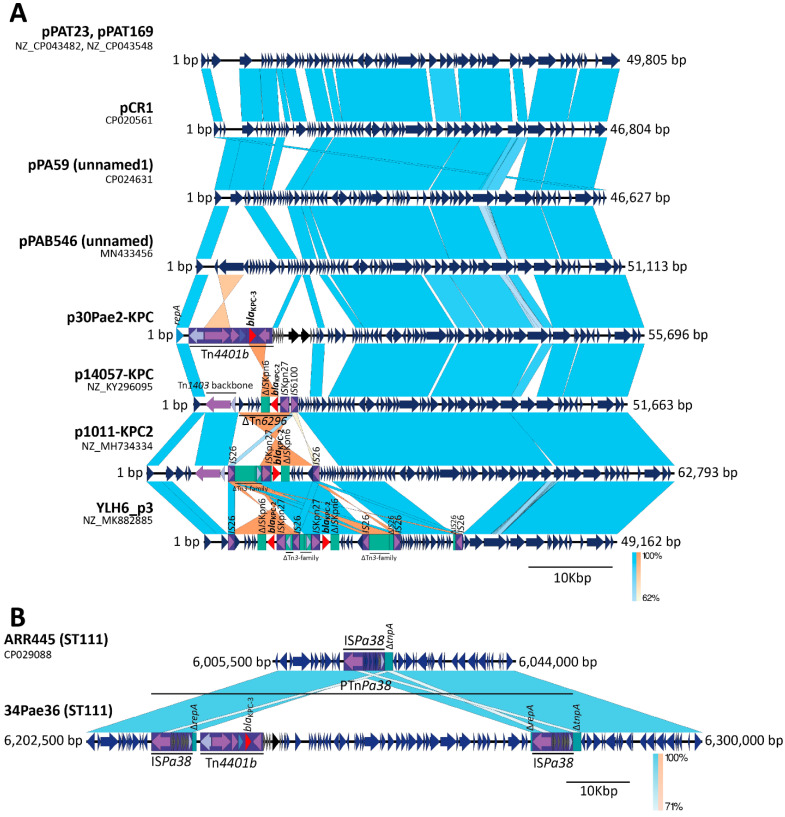
Comparative genomic analysis of the *bla*_KPC_-mobilizing platforms in *Pseudomonas aeruginosa* strains identified in this study. (**A**) Comparison of the p30Pae2-KPC with related plasmids publicly available. (**B**) Comparison of the site insertion for the plasmid p30Pae2-KPC in the isolates 30Pae2 and 34Pae36. Notably, in 30Pae2, one copy of the IS*Pa38*/Tn*Pa38* was present while in the 34Pae36 two copies were found, suggesting the possible participation of this IS in the chromosomal insertion of the plasmid. The shaded area between the sequences delimits the alignment regions with a percentage identity ≥ 62%. The red, purple, lilac, and blue arrows indicate the *bla*_KPC_ gene, transposases, resolvases, and other open reading frames, respectively. The mobile genetic elements (ISs and transposons) are highlighted and delimited by a purple rectangle. The graph has a scale line of 10,000 bp.

**Figure 5 antibiotics-14-00947-f005:**
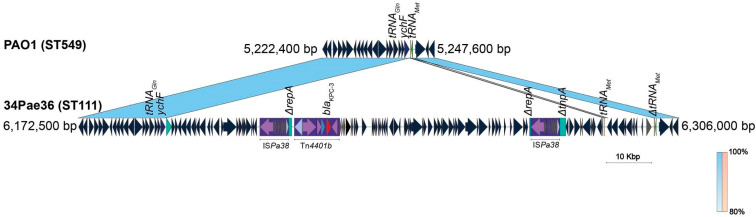
Analysis of the chromosomal IS*Pa38*/Tn*Pa38* insertion site in the genomic island region of *Pseudomonas aeruginosa* isolate 34Pae36. The localization of the different tRNAs found in this region are showed, highlighting the copy of the *tRNA^Met^* gene where the Pf1-like phage was inserted. The *P. aeruginosa* PAO1 genome was used as reference (NC_002516). The shaded area between the sequences delimits the alignment regions with a percentage identity ≥ 80%. The arrows indicate the open reading frames (blue), transposases (purple), resolvases (lilac), and the *bla*_KPC_ gene (red). The ISs and transposons are highlighted and delimited by a purple rectangle. The graph has a scale line of 10,000 bp. ST: Sequence Typing.

**Figure 6 antibiotics-14-00947-f006:**
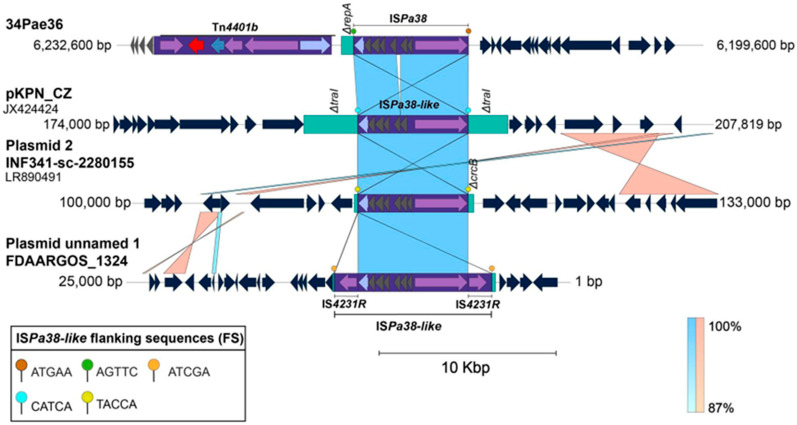
Comparative genomic analysis of the IS*Pa38*/Tn*Pa38* identified in *Pseudomonas aeruginosa* strains respect to those ISPa38/Tn*Pa38*-like elements found within *Klebsiella pneumoniae* strains. The results indicate that this IS/transposon seems to be more active in *K. pneumoniae*. The shaded area between the sequences delimits the alignment regions with a percentage identity ≥ 87%. The arrows indicate the open reading frames (blue), transposases (purple), resolvases (lilac), and the *bla*_KPC_ gene (red). The ISs and transposons are highlighted and delimited by a purple rectangle. The flanking sequences (FS) and TSDs (Target Site Duplication) for each mobile genetic element are displayed in the bottom box and are illustrated as color pallets. The graph has a scale line of 10,000 bp.

**Table 1 antibiotics-14-00947-t001:** Analysis of the IS*Pa38*/Tn*Pa38* acquisition events in 13 *Pseudomonas aeruginosa* genomes. The comparison of their flanking sequences (each one highlighted with a different background color) indicated that it has arrived at this bacterium in at least three independent acquisition events.

Insertion	Strain	Flanking Sequences		Chromosomal Position	Insertion Site	Genetic Element	Sequence Type
		Left	Right				
**1**	34Pae36 and 30Pae2	ATGAA	AGTTC	6,273,055–6,279,529	Intergenic region of IPC90_06220 and *ΔtnpA* *	Prophage	ST111
		AGTTC	TTCCA	6,212,818–6,219,281			
	AG1	ATGAA	TTCCA	6,049,321–6,055,784	Intergenic region of IPC90_06220 and *ΔtnpA*	Prophage	ST111
	AR445	ATGAA	TTCCA	6,016,755–6,022,321	Intergenic region of IPC90_06220 and *ΔtnpA*	Prophage	ST111
	Carb01	ATGAA	TTCCA	6,373,687–6,381,055	Intergenic region of IPC90_06220 and *ΔtnpA*	Prophage	ST111
	PA38182	ATGAA	TTCCA	6,623,388–6,616,919	Intergenic region of IPC90_06220 and *ΔtnpA*	Prophage	ST111
	Pa5486	ATGAA	TTCCA	6,080,366–6,086,830	Intergenic region of IPC90_06220 and *ΔtnpA*	Prophage	ST111
	PaAI2	ATGAA	TTCCA	6,054,509–6,060,982	Intergenic region of IPC90_06220 and *ΔtnpA*	Prophage	ST111
	Y82 *	ATGAA	TTCCA	5,102,848–5,109,262	Intergenic region of IPC90_06220 and *ΔtnpA*	Prophage	ST111
	RIVM-EMC2982	ATGAA	TTCCA	1,117,512–1,123,981	Intergenic region of IPC90_06220 and *ΔtnpA*	Prophage	ST111
**2**	AR_0440 **	-	AGGTA	–4,250,295	Intergenic region	Genomic Island	ST357
**3**	CF39S	AGGTA	TACCA	3,334,102–3,340,563	Intergenic region	Pathogenicity Island	ST175
	PcyII-29	AGGTA	TACC–A	3,181,009–3,187,470	Intergenic region	Pathogenicity Island	ST175
	DK2	AGGTA	TACCA	3,066,754–3,073,219	Intergenic region	Pathogenicity Island	ST386
	PABCH0	AGGTA	TACCA	3,046,061–3,412,525	Intergenic region	Pathogenicity Island	ST155

* *ΔtnpA*: Truncated transposase; ** IS*Pa38*/Tn*Pa38* is truncated at the 5′ side.

## Data Availability

The data presented in this study are available within the article. Raw data supporting this study are available from the corresponding author upon reasonable request.

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
