# Peer review of "New Insights in *bla*_KPC_ Gene Mobilization in *Pseudomonas aeruginosa*: Acquisition of *bla*_KPC-3_ and Identification of a New Tn*2*-like NTE Mobilizing *bla*_KPC-2"

_antibiotics, 2025, doi:10.3390/antibiotics14090947_

Round 1
Reviewer 1 Report
Comments and Suggestions for Authors
This manuscript performed a comparative genomic analysis of two new genetic platforms mobilizing the blaKPC-2 and blaKPC-3 in two ST111 and ST235 pandemic clones of P. aeruginosa and gave new insights of blaKPC gene mobilization. In conclusion, the authors demonstrated that the pandemic ST111 and ST235 of P. aeruginosa continue being the getters of the blaKPC gene through different mobilization routes, the jumping of conventional Tn4401b and a new Tn2-derived NTE, which were inserted in diverse plasmids. However, the authors only profiled the genetic relationship among different bacterial isolates and neither provided how blaKPC gene will affect the bacterial drug-resistance, such as related pathways or target proteins nor gave suggestions on how to prevent the transmission of the genes. Therefore, the novelty and significance of this manuscript is below average and did not meet the qualifications of this journal.
Comments on the Quality of English LanguageThis manuscript performed a comparative genomic analysis of two new genetic platforms mobilizing the blaKPC-2 and blaKPC-3 in two ST111 and ST235 pandemic clones of P. aeruginosa and gave new insights of blaKPC gene mobilization. In conclusion, the authors demonstrated that the pandemic ST111 and ST235 of P. aeruginosa continue being the getters of the blaKPC gene through different mobilization routes, the jumping of conventional Tn4401b and a new Tn2-derived NTE, which were inserted in diverse plasmids. However, the authors only profiled the genetic relationship among different bacterial isolates and neither provided how blaKPC gene will affect the bacterial drug-resistance, such as related pathways or target proteins nor gave suggestions on how to prevent the transmission of the genes. Therefore, the novelty and significance of this manuscript is below average and did not meet the qualifications of this journal.
Author Response
Dear reviewers, we appreciate your comments. We believe our article provides relevant information on how KPC is being acquired for Pseudomonas aeruginosa, identifying new genetic platforms of gene mobilization. In addition, our results show that the pandemic clones ST111 and ST235 continue to be successful catchers and maintainers of resistance genes, including KPC. Both, the third-generation sequencing and our robust comparative genomic analysis have allowed to unravel the intricate transposition events related to KPC movements.
We have made the adjustments required for you to improve our manuscript.
We added a graphical abstract for better understanding of the mobile genetic elements reported in this study.
This manuscript performed a comparative genomic analysis of two new genetic platforms mobilizing the blaKPC-2 and blaKPC-3 in two ST111 and ST235 pandemic clones of P. aeruginosa and gave new insights of blaKPC gene mobilization. In conclusion, the authors demonstrated that the pandemic ST111 and ST235 of P. aeruginosa continue being the getters of the blaKPC gene through different mobilization routes, the jumping of conventional Tn4401b and a new Tn2-derived NTE, which were inserted in diverse plasmids.
However, the authors only profiled the genetic relationship among different bacterial isolates and neither provided how blaKPC gene will affect the bacterial drug-resistance, such as related pathways or target proteins nor gave suggestions on how to prevent the transmission of the genes. Therefore, the novelty and significance of this manuscript is below average and did not meet the qualifications of this journal.
Response: We consider that our article is relevant to the scope of the journal and provides new knowledge on the dissemination of blaKPC. We have also been a reference group for the classification and discovery of new mobile genetic elements NTEKPC, as can be seen in the interactive map that we designed with the classification and update of NTEKPC elements at an international level https://www.mdpi.com/article/10.3390/antibiotics12040658/s1 or with the discover of Tn6454 in Klebsiella pneumoniae. Therefore the inclusion of the new NTEKPC-IIg in ST111 is relevant for the dissemination of blaKPC in Pseudomonas aeruginosa, additionally, the identification of blaKPC-3 in ST235 is not a frequent event, this being the second Colombian report, added to the extensive comparative genomics carried out.
Reviewer 2 Report
Comments and Suggestions for Authors
The manuscript titled "New insights in blaKPC gene mobilization in Pseudomonas aeruginosa: Acquisition of blaKPC-3 and identification of a new Tn2-2 like NTE mobilizing blaKPC-2." presents relevant and innovative data regarding the resistance of certain pathogens to antibiotics, a critical issue for public health. However, a more thorough review by the authors is necessary, including some modifications to enhance the clarity and accuracy of the presented information:
Lines 49-50: The authors should rewrite this sentence to ensure greater clarity and accuracy in conveying the information.
Line 56: What does the term Ukraine in parentheses mean?
Line 93: How was the genomic DNA extracted? It is important to detail the methodology used to ensure the reproducibility of the results.
Line 95: Although it is well known that PFGE is performed using an agarose gel, it would be beneficial for the authors to specify the concentrations and/or volumes of the reagents used in the assay. This level of detail would enhance the transparency and reproducibility of the study.
Line 130: Since this is the first supplementary table mentioned in the manuscript, the authors should designate it as Table S1 and, from there, follow a logical order of citation throughout the text.
Line 188: Figure 1, currently placed between lines 209-218 (including the figure and its caption), should be relocated starting from line 188, as the corresponding section of the text appears between lines 182-187. This adjustment would improve readability and ensure a clearer connection between the textual content and the illustration.
Lines 190-193: This sentence of the results section should be moved to the discussion section rather than the results section.
Furthermore, it would be important for the authors to review the results section, as many passages actually belong in the discussion section. This restructuring would not only facilitate reading but also enrich the critical analysis of the work, ensuring that the results are presented clearly and objectively, while interpretations are properly discussed. Some of the passages that need to be moved to the discussion section are found between lines 280-284, lines 289-292, lines 297-300 and lines 322-330. This adjustment is essential to ensure that the analysis and interpretation of the data are placed in the appropriate section, contributing to a more cohesive and logical structure of the manuscript.
The references section should be reviewed as it is currently disorganized.
Line 597: Reference 55 is not identified in the text.
Line 598: Figure 4 is placed at the end of the document inappropriately. Ideally, and if it is possible, figures should be positioned immediately after the sections of the text they refer to, which is the case of figure 4.
Line 631: Table S2 is mistakenly listed as reference 56. However, supplementary tables should not be treated as bibliographic references, as they are part of the manuscript itself. It would be appropriate to correct this citation by treating the table as a section of the manuscript rather than an external reference.
Table S3: What is the size of the DNA fragment generated by these primers?
Author Response
Dear reviewers, we appreciate your comments. We believe our article provides relevant information on how KPC is being acquired for Pseudomonas aeruginosa, identifying new genetic platforms of gene mobilization. In addition, our results show that the pandemic clones ST111 and ST235 continue to be successful catchers and maintainers of resistance genes, including KPC. Both, the third-generation sequencing and our robust comparative genomic analysis have allowed to unravel the intricate transposition events related to KPC movements.
We have made the adjustments required for you to improve our manuscript.
We added a graphical abstract for better understanding of the mobile genetic elements reported in this study.
Comment 1: The manuscript titled "New insights in blaKPC gene mobilization in Pseudomonas aeruginosa: Acquisition of blaKPC-3 and identification of a new Tn2-2 like NTE mobilizing blaKPC-2." presents relevant and innovative data regarding the resistance of certain pathogens to antibiotics, a critical issue for public health. However, a more thorough review by the authors is necessary, including some modifications to enhance the clarity and accuracy of the presented information:
Lines 49-50: The authors should rewrite this sentence to ensure greater clarity and accuracy in conveying the information.
Response: The sentence was changed to: “Our results reveal that the pandemic high-risk clones ST111 and ST235 of P. aeruginosa continues to spread blaKPC gene through different mobile genetic elements, jumping of conventional Tn4401b and acquiring new Tn2-derived NTE, which were inserted in diverse plasmids”
Comment 2: Line 56: What does the term Ukraine in parentheses mean?
Response: The highest rate of carbapenem resistance in Europe was reported in Ukranie, the aclaration was added in the line 86 “(reported in Ukraine)”.
Comment 3: Line 93: How was the genomic DNA extracted? It is important to detail the methodology used to ensure the reproducibility of the results.
Response: The sentence “The genomic DNA of isolates was extracted with DNeasy UltraClean Microbial Kit (QIAGEN)” was added in line 122 to “source of the bacterial isolates, susceptibility profiles and molecular detection of carbapenemases” section
Comment 4: Line 95: Although it is well known that PFGE is performed using an agarose gel, it would be beneficial for the authors to specify the concentrations and/or volumes of the reagents used in the assay. This level of detail would enhance the transparency and reproducibility of the study.
Response: The PFGE assay was described in the lines 127-144 “Genetic relationship was determined by pulsed-field electrophoresis (PFGE) analysis, following the Centers for Disease Control and Prevention (CDC) with minor modifications [1]. P. aeruginosa isolates were cultivate in BHI broth and incubated at 37°C with shaking at 200 rpm for 12 hours, to an OD625 of 1.5 in TE buffer (100 mM Tris + 100 mM EDTA, pH 8.0). 1% blocks were prepared with SeaKem® Gold Agarose, with150 µL of bacterial suspension and 7.5 µL of proteinase K (20 mg/mL). For lysis, 1 ml of lysis buffer (50 mM Tris, 50 mM EDTA, pH 8.0 + 1% Sarcosyl) with 5 µl of proteinase K (20 mg/mL) was added to the blocks and incubated for 2 hours at 55°C at 175 rpm. Two washes were then performed with sterile deionized distilled water at 55°C and two washes with TE buffer (10 mM Tris, 1 mM EDTA, pH 8.0). The blocks were stored at 4°C until use. Restriction was performed with 12 units of the restriction enzyme SpeI (10 U/µL: Promega) in 1X Buffer B for 5 hours at 37°C. Restriction fragments were separated on a 1% SeaKem® Gold Agarose gel in 0.5X TBE (Tris base; boric acid; 0.5M EDTA (pH 8.0); dH2O) buffer using the CHEFFII equipment (BioRad) with the following running conditions: initial time of 6.8 seconds and final time of 35.4 seconds for 23 hours at an angle of 120° and a voltage of 6.0. The gels were stained with ethidium bromide (10 mg/mL). The image was captured with the ChemiDoc™ MP Imaging System (BioRad) and the dendrogram was obtained using the GelCompar II program (Applied Maths NV). For the analysis, Salmonella serotype Braenderup H9812 was used as a normalizing strain and the pulsotypes were defined accourding to the criteria of Tenover et al. with a cut-off point greater than 80% similarity [2].”
Comment 5: Line 130: Since this is the first supplementary table mentioned in the manuscript, the authors should designate it as Table S1 and, from there, follow a logical order of citation throughout the text.
Response: Supplementary tables order was adjusted at line 181, 200 and 231.
Comment 6: Line 188: Figure 1, currently placed between lines 209-218 (including the figure and its caption), should be relocated starting from line 188, as the corresponding section of the text appears between lines 182-187. This adjustment would improve readability and ensure a clearer connection between the textual content and the illustration.
Response: Figure 1 was relocated in line 260 of the document.
Comment 7: Lines 190-193: This sentence of the results section should be moved to the discussion section rather than the results section.
Response: The sentence “These STs have already been previously reported with blaKPC gene in Colombia [3–7] and other countries such as Brazil and Germany [8,9] “ was added to line 488
Comment 8: Furthermore, it would be important for the authors to review the results section, as many passages actually belong in the discussion section. This restructuring would not only facilitate reading but also enrich the critical analysis of the work, ensuring that the results are presented clearly and objectively, while interpretations are properly discussed. Some of the passages that need to be moved to the discussion section are found between lines 280-284, lines 289-292, lines 297-300 and lines 322-330. This adjustment is essential to ensure that the analysis and interpretation of the data are placed in the appropriate section, contributing to a more cohesive and logical structure of the manuscript.
Response: Sentence of lines 280 – 284 “These results show that the chromosomal blaKPC-3 insertion in the isolate 34Pae36 was not produced for the Tn4401b transposition, as it was previously reported by us in other Colombian isolates [4], but for plasmid insertion. Now, what possible genetic mechanism could be involved?” were not moved because we consider that are necessary for the close and connection between the sections. Now are in lines 336 and 339.
Sentence of lines 289 – 292 “The information about this IS is scarce. According to the ISFinder database, the ISPa38 was reported only until 2011 using an automatic “in silico” search within the P. aeruginosa strain DK2, a highly frequent clone causing infections in patients with cystic fibrosis [10].“ was removed.
Sentence of lines 297– 300 “This suggests that the chromosomal insertion of the plasmid was accompanied by a duplication of this IS” was not removed because provides to readers a possible explanation in a clear and concrete way to continue with the results of insertion analysis, now in lines 382-383.
Sentence of lines 322-330 “Then, how could P. aeruginosa have acquired the ISPa38/TnPa38? From our analysis there are several points to highlight, first, the ISPa38/TnPa38 is almost exclusive to P. aeruginosa; when we examined the frequency of this IS within Pseudomonas spp, we found that it has only been described in P. aeruginosa strains (except for the clinical Pseudomonas putida strain H8234) [11]. Second, the ISPa38/TnPa38 is scarce in P. aeruginosa, from 5,678 genomes publicly available from NCBI (last accession November 2021), this IS was only found in 13 genomes.” were not modified due the frequency of ISPa38/TnPa38 in Pseudomonas spp. and P. aeruginosa described in the text and table 1 are necessary in the results sections to showing the ISPa38/TnPa38 acquisition events with their flanking sequences and insertion sites, these are now in lines 404-409.
Comment 9: The references section should be reviewed as it is currently disorganized.
Response: It appears that a number of additional references were generated to the final format that were not contemplated in the document.
Comment 10: Line 597: Reference 55 is not identified in the text.
Response: Reference 54 was the last one in the previous version, now is 62
Comment 11: Line 598: Figure 4 is placed at the end of the document inappropriately. Ideally, and if it is possible, figures should be positioned immediately after the sections of the text they refer to, which is the case of figure 4.
Response: All tables and figures were organized next to each section.
Figure 1, Line 260.
Figure 2, line 268.
Figure 3, line 318.
Figure 4, line 340.
Figure 5, line 432.
Table 1, line 439.
Figure 6, line 444.
Comment 12: Line 631: Table S2 is mistakenly listed as reference 56. However, supplementary tables should not be treated as bibliographic references, as they are part of the manuscript itself. It would be appropriate to correct this citation by treating the table as a section of the manuscript rather than an external reference.
Response: In our last version the last reference number was 54. We sent the document with this reference revision.
Comment 13: Table S3: What is the size of the DNA fragment generated by these primers?
Response: Amplicon sizes were added to the Table S3
Reviewer 3 Report
Comments and Suggestions for Authors
The manuscript by Abril et al. investigated the genomic mechanisms of carbapenem resistance in Pseudomonas aeruginosa, a major healthcare-associated pathogen. Their study identified two key resistance strategies: (1) blaKPC-2 mobilization via a Tn2-derived platform in ST235 and (2) blaKPC-3 acquisition in ST111 through plasmid and chromosomal insertion via ISPa38 (proposed as TnPa38). These findings highlight the dynamic evolution of blaKPC in P. aeruginosa and the need for ongoing surveillance.
Overall, this paper highlighted the diverse genetic strategies driving blaKPC mobilization in P. aeruginosa, underscoring the need for continued genomic surveillance of antibiotic resistance. However, the authors do not provide sufficient details in the background regarding the significance of their findings. Taking these points and the comments below into consideration, we recommend the manuscript for publication in your journal following minor revisions:
- This article proposed the novel finding in the genomic mechanisms of carbapenem resistance in Pseudomonas aeruginosa. What strategies can be implemented in hospital environments to reduce antibiotic pressure and thus slow the selection for resistance mechanisms?
- In the discussion, the author mentioned the shift from blaVIM to blaKPC dominance in Colombian isolates. Considering that, what factors might have driven this epidemiological change?
- What infection control measures could be informed by the discovery of these novel blaKPC mobilization routes? It would be better if the author could provide some discussion on the application point in the discussion part.
Author Response
Dear reviewers, we appreciate your comments. We believe our article provides relevant information on how KPC is being acquired for Pseudomonas aeruginosa, identifying new genetic platforms of gene mobilization. In addition, our results show that the pandemic clones ST111 and ST235 continue to be successful catchers and maintainers of resistance genes, including KPC. Both, the third-generation sequencing and our robust comparative genomic analysis have allowed to unravel the intricate transposition events related to KPC movements.
We have made the adjustments required for you to improve our manuscript.
We added a graphical abstract for better understanding of the mobile genetic elements reported in this study.
Comment: The manuscript by Abril et al. investigated the genomic mechanisms of carbapenem resistance in Pseudomonas aeruginosa, a major healthcare-associated pathogen. Their study identified two key resistance strategies: (1) blaKPC-2 mobilization via a Tn2-derived platform in ST235 and (2) blaKPC-3 acquisition in ST111 through plasmid and chromosomal insertion via ISPa38 (proposed as TnPa38). These findings highlight the dynamic evolution of blaKPC in P. aeruginosa and the need for ongoing surveillance.
Overall, this paper highlighted the diverse genetic strategies driving blaKPC mobilization in P. aeruginosa, underscoring the need for continued genomic surveillance of antibiotic resistance. However, the authors do not provide sufficient details in the background regarding the significance of their findings. Taking these points and the comments below into consideration, we recommend the manuscript for publication in your journal following minor revisions:
Comment 1: This article proposed the novel finding in the genomic mechanisms of carbapenem resistance in Pseudomonas aeruginosa. What strategies can be implemented in hospital environments to reduce antibiotic pressure and thus slow the selection for resistance mechanisms?
Response: Antimicrobial resistance (AMR), particularly antibiotic resistance, poses a complex global challenge requiring coordinated, multidisciplinary efforts at local, national, and international levels. Strong political commitment is essential for developing evidence-based policies, implementing regulations, and controlling inappropriate antibiotic use and promotion in human and animal health. Novel strategies, including resistant gene inactivation and alternative approaches that do not induce resistance, are under investigation. However, their long-term efficacy remains to be fully evaluated, with ongoing contributions from regulatory bodies and institutions.
The following table outlines the primary strategies that can be implemented in hospitals to reduce antimicrobial resistance (AMR) and were added in discussion section (see attached file):
Comment 2: In the discussion, the author mentioned the shift from blaVIM to blaKPC dominance in Colombian isolates. Considering that, what factors might have driven this epidemiological change?
Response: The discussion section in lines 517 to 553 was modified to “In the present study, our results show that, in the six health institutions, the blaKPC isolates have displaced the blaVIM isolates and are already the most prevalent, suggesting a successful adaption of this resistance mechanism to this non-fermenting bacterium in our country. The blaKPC gene has become the predominant carbapenemase-encoding gene in Colombia, largely due to its efficient dissemination through mobile genetic elements. This increasing frequency of KPC-producing P. aeruginosa isolates can be related with the mobilization of the blaKPC gene toward new plasmid backbones, some of them, with an environmental origin; coming from blaKPC-positive plasmids circulating in Klebsiella pneumoniae, where this gene circulates with very high frequency and within diverse and very variables genetic platforms (including MGE and plasmids), these MGEs could be consulted at https://maphub.net/LGMB/KPC-Pseudomonas-aeruginosa-LGMB [12,13].
Studies have shown that blaKPC-2 and blaKPC-3 are the most common variants, frequently detected in healthcare-associated infections (HAIs) and occasionally in community-acquired infections [14,15]. Regarding blaKPC-3, our findings were even more interesting, showing that this gene is been mobilized within a little-known plasmid structure, previously detected in isolates of environmental origin in Asia, some of these with blaKPC incorporated, but variant 2, and mobilized for no classical Tn4401 (Figure 2). In contrast, in the case of the plasmid p30Pae2-KPC, the Tn4401b was probably the first element responsible to mobilize the gene, but the insertion into the plasmid was helped maybe for another mobile structure (no TSD was found in Tn4401b). In this case, we also suppose the blaKPC-3 gene has arrived at P. aeruginosa from K. pneumoniae through the shared plasmids, due to that, in Colombia, the blaKPC-3 is the more frequently variant detected in Klebsiella pneumoniae isolates [55,56].
The overuse and misuse of carbapenems in Colombian healthcare settings have exerted strong selective pressure, favoring the emergence and dissemination of blaKPC-producing bacteria. Studies have shown that prior exposure to carbapenems is a significant risk factor for infections with carbapenem-resistant Enterobacterales (CRE) [44,57]. Deficiencies in infection control practices have played a pivotal role in the spread of blaKPC-producing bacteria in Colombian healthcare facilities. Horizontal gene transfer and clonal dissemination have been facilitated by inadequate hand hygiene, improper sterilization of medical devices, and insufficient isolation of infected patients [58]. These lapses have allowed blaKPC-producing isolates to thrive in hospital environments, contributing to their dominance over blaVIM-producing isolates. To reduce antibiotic pressure and thus slow the selection for resistance mechanisms in P. aeruginosa clinical isolates and other Enterobacterales species, it is essencial that primary strategies can be implemented in hospitals (table S4) and more genomic studies that allowing to know the mobile genetic elements with resistant determinants that are spreading in the institution, that allow a better dissemination control.”
Comment 3: What infection control measures could be informed by the discovery of these novel blaKPC mobilization routes? It would be better if the author could provide some discussion on the application point in the discussion part.
Response: The discussion section in lines 553 to 566 was modified to ”The implementation of regional genomic surveillance programs has been shown to be effective in tracking the spread of blaKPC-harboring pathogens. For example, a study in three U.S. states demonstrated that genomic surveillance could identify distinct transmission pathways, such as importation versus clonal expansion, and inform targeted interventions [20] The clinical implementation of WGS has been shown to be feasible and effective for identifying transmission events of multidrug-resistant pathogens, including those harboring blaKPC. For example, a study in three Australian hospitals demonstrated that WGS-based surveillance can identify clusters of multidrug-resistant organisms and inform infection control interventions [21].
Standard and contact infection prevention and control practices, including hand hygiene, patient isolation, and environmental disinfection, remain critical for preventing the spread of blaKPC-harboring pathogens. Studies have shown that these practices, when implemented consistently, can reduce the transmission of multidrug-resistant organisms in healthcare settings [22,23]”.

Round 2
Reviewer 1 Report
Comments and Suggestions for Authors
The authors have revised the manuscript accordingly.